# Estimating Optimal Context Length for Hybrid Retrieval-augmented Multi-document Summarization

**Adithya Pratapa**     **Teruko Mitamura**
Language Technologies Institute
Carnegie Mellon University
{vpratapa, teruko}@cs.cmu.edu

## Abstract

Recent advances in long-context reasoning abilities of language models led to interesting applications in large-scale multi-document summarization. However, prior work has shown that these long-context models are not effective at their claimed context windows. To this end, retrieval-augmented systems provide an efficient and effective alternative. However, their performance can be highly sensitive to the choice of retrieval context length. In this work, we present a hybrid method that combines retrieval-augmented systems with long-context windows supported by recent language models. Our method first estimates the optimal retrieval length as a function of the retriever, summarizer, and dataset. On a randomly sampled subset of the dataset, we use a panel of LMs to generate a pool of silver references. We use these silver references to estimate the optimal context length for a given RAG system configuration. Our results on the multi-document summarization task showcase the effectiveness of our method across model classes and sizes. We compare against length estimates from strong long-context benchmarks such as RULER and HELMET. Our analysis also highlights the effectiveness of our estimation method for very long-context LMs and its generalization to new classes of LMs.[1]

## 1 Introduction

Language models increasingly support longer context windows, leading to useful applications in large-scale multi-document summarization. Recent work has shown that these models are not very effective at their claimed context windows (Hsieh et al., 2024; Yen et al., 2025). An alternative to the full context setting is retrieval-augmented generation (RAG), and previous work has illustrated its effectiveness for long input processing (Asai et al., 2024; Li et al., 2024). RAG systems facilitate better use of the LM context windows by passing only the most relevant information to the model. However, the choice of retrieval length that provides peak RAG performance is often unclear and sensitive to the choice of retriever, language model, and downstream task (Jin et al., 2025). In this work, we present a methodology for estimating this optimal retrieval length as a function of the retriever, summarizer, and dataset. In addition to providing gains over the full context setting, our method also outperforms the context-length estimates identified by standard long-context evaluation benchmarks. Figure 1 provides a schematic overview of our method.

Previous efforts to combine RAG and long-context LMs focused on query-based routing (Li et al., 2024), or iterative RAG (Yue et al., 2025). While these methods are effective, they rely on the model's ability to accurately determine the scope of information need and self-evaluate its own output. This might not always be a feasible option, especially for smaller LMs. In this work, we take a complementary approach to combine RAG and long-context and show its effectiveness for models ranging from 0.5B to 72B parameters. We evaluate on a challenging large-scale multi-document summarization dataset (Laban et al., 2024).

---

[1]Our code is publicly available at https://github.com/adithya7/hybrid-rag.

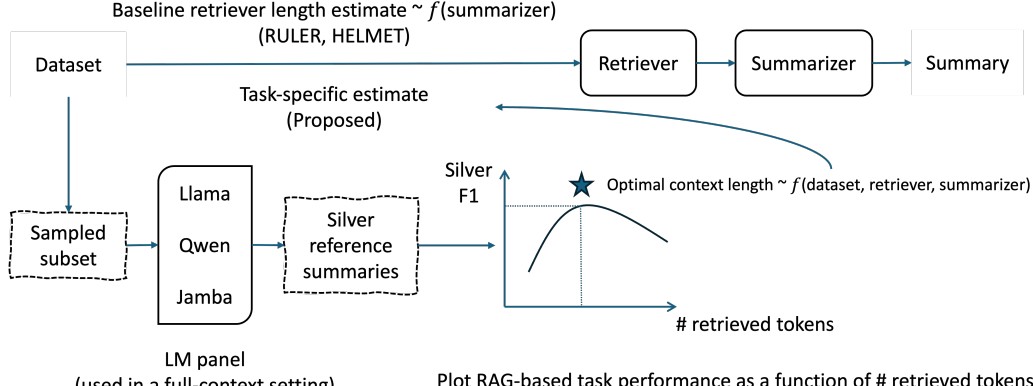

Figure 1: A schematic overview of our proposed method. Unlike traditional benchmarks, we estimate the optimal retrieval length as a function of dataset, retriever and summarizer. Given a dataset, we first sample a fraction of examples. On this subset, we run a panel of LMs in a full-context setup to create silver candidates. We then identify the top silver candidates using Minimum Bayes Risk decoding. With the help of these silver candidates, we estimate the optimal retrieval length for the given experiment config.

In a recent work, Jin et al. (2025) compared the RAG performance of varying model sizes on the question-answering task and found that the optimal retrieval length varies considerably across model sizes. They also found that this length is sensitive to the choice of retriever. Similarly, Yu et al. (2024) noted the sensitivity of optimal retrieval length to the downstream task. Based on these observations from previous work, we hypothesize that the retrieval length that provides peak performance should be modeled as a function of the three main components of the RAG pipeline: retriever, summarizer, and dataset. For our baselines, we use two popular long-context evaluation benchmarks, RULER (Hsieh et al., 2024) and HELMET (Yen et al., 2025). They benchmark models on a suite of tasks with inputs of increasing lengths. RULER focuses on synthetic retrieval tasks, while HELMET includes NLP tasks such as LongQA and summarization. Although these provide *effective* context length estimates for individual LMs, these estimates are often agnostic to the downstream dataset and the retrievers when used in the RAG setting.

Given a dataset, we first create a subset of representative examples by random sampling. We then use a panel of LMs to compile a candidate set of silver reference summaries. In our panel, we include LMs from the Qwen (Qwen et al., 2025), Llama (Grattafiori et al., 2024) and Jamba (Team et al., 2024) series. From the pool of candidate silver references, we use Minimum Bayes Risk decoding (Kumar & Byrne, 2004) to identify the top silver reference summaries. For a given combination of retriever and summarizer models, we perform a search over context lengths on this silver subset to estimate the optimal retrieval length. Unlike baseline methods, our approach is customized to the specific experiment configuration (dataset, retriever, and summarizer). Our method is based on two key observations. First, larger LMs show robust performance across a broad range of context lengths. This is mainly due to their enhanced ability to deal with noise in the retrieved input (Jin et al., 2025). Second, to identify a task-specific estimate, we can approximate the gold summaries with silver candidates sampled from strong long-context LMs.

We evaluated our method for the multi-document summarization task using the SummHay dataset (Laban et al., 2024). Our results show that all retrieval-based methods (baselines and ours) significantly outperform full-context. Our method performs the best in most settings, followed by HELMET- and RULER-based estimates. Although HELMET-based estimates sometimes perform comparable to our method, neither the LongQA nor summarization task-based HELMET estimates consistently perform better. Notably, our method performs much better on very long-context LMs such as Qwen 2.5 1M and ProLong 512k. Our analysis also shows that our method generalizes well to model classes outside of our panel (e.g.,

Phi-3). We also perform ablation experiments on our LM panel as well as the size of our sampled subset.

## 2 Estimating Optimal Context Length for Retrieval

For the multi-document summarization task, given a long input and a query, we have two possible systems. First, the entire input is fed directly into a long-context summarizer that supports such lengths (full-context). Second, we use the query to rank the documents and only pass the top-k relevant documents to the summarizer (RAG). Previous work has shown that long-context models are not effective at their claimed context windows, and RAG can help improve task performance (Yu et al., 2024; Pratapa & Mitamura, 2025).

Benchmarks such as RULER and HELMET provide a comprehensive evaluation of long-context models across a suite of NLP tasks, including QA and summarization. However, these benchmarks focus solely on the model and do not study the effects of unseen downstream datasets and the retrievers used in RAG settings. Previous work has briefly studied this with Jin et al. (2025) noting significant variance in long-context RAG performance depending on the choice of LM and retriever. Yu et al. (2024) noted similar behavior for question-answering tasks. Giorgi et al. (2023) studied the effects of retriever and summarizer for short-context open-domain multi-document summarization. Therefore, we hypothesize that the optimal context length estimate for a RAG system should be a function of the retriever, summarizer, and specific downstream task.

Our proposed method is centered on two key observations. First, large LMs show robust performance across a broad range of context lengths because of their enhanced ability to deal with noise in the retrieved input. Jin et al. (2025) observed this behavior for long input QA tasks. Second, gold references can be approximated by silver references sampled from strong long-context LMs. For a given dataset (D), retriever (R), and summarizer (S), our method involves the following steps. See Figure 1 for an overview of our method.

1. We sample a subset of the dataset (D). Each example in this subset constitutes a set of documents and a query. (§2.2)
2. We used a panel of LMs (§2.1) to generate summaries for this subset. These summaries serve as our candidate silver references. (§2.2)
3. We use Minimum Bayes Risk decoding to identify the best silver references. (§2.2)
4. We perform a search for retrieval lengths (8k to 80k) by comparing the system-generated summary (using R & S) against the silver references. This search gives us the optimal retrieval length estimate for our RAG setup. (§2.3)
5. Finally, on the full dataset, we retrieve the top-k documents that fit into this length estimate (using R) before generating a summary (using S).

### 2.1 LM panel

In our LM panel, we include a diverse class of models. Panels of diverse LMs have previously been explored for evaluation and are considered a strong alternative to a single LM evaluator (Verga et al., 2024).

**Large LMs:** We choose Qwen-2.5 72B (Qwen et al., 2025), Llama-3.3 70B (Grattafiori et al., 2024), and Jamba-1.5 Mini (Team et al., 2024). These are the largest models from each class that we could run locally.[2]

**Long-context LMs:** We include two smaller LMs that are specifically trained for long-context tasks, Qwen-2.5-1M 14B (Yang et al., 2025) and ProLong 512K (Gao et al., 2024). ProLong is continually trained on long texts starting from the Llama-3 8B model.

In our pool, we focussed on including diverse models while being within our compute budget to run these models locally. Our panel can be easily modified with newer variants of these models as well as include API-based models.

---

[2]We couldn't run Llama 405B and Jamba 1.5 Large (400B) locally on our setup.

## 2.2 Generating silver references

To begin, we select a fraction of the examples from the dataset (default: 25%) using a uniform sampling algorithm (without replacement). We run our silver reference generation on this data subset. We leave out the gold references and do not use them in our context-length estimation procedure.

**Silver references:** We run our LM panel to create a pool of candidate silver references. We used temperature sampling ($\tau = 0.5$) to generate three candidate summaries for each LM. We use LMs in a full-context setup and do not assume any optimal context length.

**Pooling:** We experiment with two ways to collect our final set of silver references. First, we used a single LM from the panel and select the three sampled candidates as our silver references. Second, we collect many candidates by pooling outputs from all LMs in our panel. In this scenario, we use Minimum Bayes Risk (MBR) decoding to identify the three highest scoring candidates. We follow previous work (Suzgun et al., 2023; Bertsch et al., 2023) to compute the similarity between each pair of candidates and obtain the alignment scores among the candidates. To be consistent with our downstream evaluation metric, we use the A3CU F1 score as our utility metric in MBR decoding.

Our use of MBR decoding here borrows ideas from previous summarization works, specifically post-ensemble (Kobayashi, 2018) and crowd sampling (Suzgun et al., 2023). Similarly to Kobayashi (2018), we use a model ensemble in the post-processing stage. We follow Suzgun et al. (2023) to use temperature sampling and a neural utility metric. However, our utility metric differs from the BLEURT and BERTScore used in Suzgun et al. (2023).

## 2.3 Search for optimal retrieval length

To identify the optimal length for the retrieval step, we search a wide spectrum of context lengths from 8K to 80K tokens in 8K intervals. For each context length C, we run the RAG pipeline on the silver dataset by retrieving up to C tokens (see §3.2) and passing them to the summarizer. We evaluate the system generated summaries against the silver references. We generate three predictions per example using temperature sampling ($\tau = 0.5$) and take the average A3CU F1 score (see §3.1). For efficiency reasons, we choose the smallest context length that falls within a standard deviation of the maximum score as our optimal context length. Previous works RULER and HELMET use coarser intervals for context lengths (multiples of 8K).

Yue et al. (2025) is closely related to our work. For the long input question answering task, they propose an iterative RAG method that uses inference-time scaling. Unlike traditional RAG, their method iteratively generates subqueries and retrieves additional documents as needed before generating the final answer. They present a computation allocation model that optimizes task performance based on three parameters: number of documents, number of demonstrations, and maximum number of iterations. Our setting differs considerably from this work. For multi-document summarization task, we have a fixed set of documents, and including demonstrations in the prompt is often infeasible. We believe that our single-step retrieval solution can be combined with such iterative methods to further improve task performance. We leave this extension to future work.

## 3 Experimental Setup

In this section, we describe our dataset, the evaluation metric, baselines, and the systems used for the retrieval and summarization tasks.

### 3.1 Dataset & Metric

**SummHay:** Proposed by Laban et al. (2024), this is a multi-document summarization curated using GPT-3.5 and GPT-4o, starting with summary insights followed by document generation. Each input typically consists of 100 documents (avg. length 884 words), and the

summary consists of an average of 185 words. This dataset includes 92 examples that cover the news and conversational domains.

**Metric:** For the summarization task, we report the F1 score of the reference-based Atomic Content Unit (A3CU) metric (Liu et al., 2023b). This model-based metric is trained to predict a score that measures the overlap of atomic content units (Liu et al., 2023a) between the system-generated and reference summaries. Previous work has found that this metric is strongly correlated with human evaluation for both single (Liu et al., 2023b) and multi-document summarization (Pratapa & Mitamura, 2025).

## 3.2 Retrieval systems

For our retrieval task, we use entire documents as retrieval units and obtain document embeddings using Qwen-2-based GTE models (Li et al., 2023). We then compute cosine similarity between document and query embeddings and pick the top-k documents that fit within the specified context length.

Jin et al. (2025) analyzed the effect of the retriever on optimal context lengths in RAG settings and found that the stronger retriever has shorter optimal lengths than the weaker retrievers. To see the impact of this in our setting, we experiment with two sizes of GTE embeddings, Qwen-2-1.5B[3] and Qwen-2-7B.[4]

We acknowledge the impact of chunking strategies on RAG performance (Chen et al., 2024), however, shorter chunks might need additional recontextualization.[5] We leave the exploration of fine-grained chunking strategies to future work.

## 3.3 Summarization systems

For the summarization task, we use the instruction fine-tuned variants from Qwen-2.5, Llama-3, ProLong, and Phi-3 series of models.

**Qwen-2.5:** We experiment with multiple sizes from this series including 0.5B, 1.5B, 3B, 7B, 14B, 32B, and 72B (Qwen et al., 2025). The smaller models ($\leq$3B) only support a context length of 32K, while the larger models support up to 128K tokens. For the smaller models, we report RAG @ 32K as their full-context performance.

**Qwen-2.5-1M:** These are long-context variants of the Qwen-2.5 7B and 14B models (Yang et al., 2025) supporting up to a context length of 1M tokens.

**Llama-3**: We include 1B, 3B, 8B and 70B models in our experiments. All models support a context length of 128K tokens (Grattafiori et al., 2024).

**ProLong:** Gao et al. (2024) continually fine-tuned Llama-3-8B-Instruct on long texts of up to 512K tokens. They are first trained on 20B training tokens of 64K data, followed by another 20B training tokens of 512K data. We experiment with the 64K and 512K variants.

**Phi-3:** We use three model sizes, Mini (3.8B), Small (7B) and Medium (14B). All of these models support context lengths of up to 128K tokens (Abdin et al., 2024).

We use vLLM (Kwon et al., 2023) for our inference runs, using up to four 48GB L40S GPUs in our experiments. For each set of input documents, we sample three summaries using temperature sampling ($\tau = 0.5$). To provide a fair comparison of our systems, we limit all of our inputs to a maximum of 128K tokens. See Appendix §A.2 for additional details about our task prompts, tokenization, truncation strategies, and summary lengths.

## 3.4 Baselines

**Full-context**: In this setup, we utilize the full context window supported by the summarization model. Typically, larger models also tend to perform well in long-context tasks. To

---

[3] https://huggingface.co/Alibaba-NLP/gte-Qwen2-1.5B-instruct
[4] https://huggingface.co/Alibaba-NLP/gte-Qwen2-7B-instruct
[5] https://www.anthropic.com/news/contextual-retrieval

study this behavior, we include models of varying sizes in our experiments. Inputs longer than the supported context window are truncated starting with the longest documents.

For our RAG baselines, we rely on widely used long-context benchmarks RULER and HELMET that estimate efficient context windows for language models. For these baselines, we limit the number of tokens retrieved to an efficient context window of the corresponding summarization model.

**RULER** (Hsieh et al., 2024) benchmark consists of a collection of synthetic retrieval tasks at varying input lengths (8K, 16K, 32K, 64K and 128K). For a given LM, this benchmark evaluates its retrieval performance at these input lengths and determines an effective context window by using the performance of Llama-2-7B @ 4K as the threshold. We used the effective context windows reported in previous work as our baseline estimates.

**HELMET** (Yen et al., 2025) benchmark covers a suite of NLP tasks, with multiple datasets included in each task. The tasks are recall, RAG, citation, re-ranking, ICL, LongQA, and summarization. For each dataset, they evaluate system performance at varying input lengths (same set as RULER). They report task averages, as well as a HELMET average. As our baseline, we select the two most relevant subtasks, LongQA and summarization. For each task and LM, we choose the context length with the highest task average as the effective context window for LM.

Note that both RULER and HELMET benchmarks evaluate model in a full-context setting but often find the optimal context window to be much lower than the claimed (or supported) context window by the LM. In our experiments, we used previously reported scores on the RULER and HELMET benchmarks. See Table 7 in Appendix §A.1 for a full list of context length estimates from our baselines.

## 4   Results

In Table 1, we compare our method with the baselines on the SummHay dataset. All RAG-based systems (baselines and ours) outperform full-context setup. Our method consistently shows strong performance across model classes, sizes, and retrievers. Although the RULER- or HELMET-based estimates do well in specific instances, neither is consistently better across all settings. Among our baselines, we find that the LongQA-based estimate from HELMET performs the best. In Table 8 in the Appendix, we report the context window estimates used in each experiment setting as well as the standard deviation across three random seeds.

## 5   Discussion & Analysis

We now analyze the effectiveness of our method in various settings. In §5.1, we look at very long context LMs (>500K). In §5.2, we evaluate the generalization of our estimation method to a model class not included in our LM panel. In §5.3, we contrast our pooled estimate with those obtained using silver references from a single large LM. We also evaluate the effect of the dataset sampling ratio on the quality of the estimated context length (§5.4). Finally, in §5.5, we discuss the performance and efficiency gains with our RAG setup.

### 5.1   Very long-context LMs

As LMs improve their long-context reasoning, there is often a reduced need for RAG. Recent work (Yu et al., 2024) argues for the combination of long-context models and RAG, and our results in Table 1 reinforce this argument. However, we want to test the effectiveness of our method on LMs carefully trained for long-context reasoning. For this analysis, we chose Qwen 2.5 1M models (Yang et al., 2025) (7B, 14B), and ProLong 512K (Gao et al., 2024). These models are continually trained on long texts and show almost perfect performance at 128K context length on HELMET. We report results in Table 2. Our method consistently outperforms the baselines. We leave the exploration of closed-weight API-based models such as Gemini 1.5 Pro to future work.

| Retriever | Summarizer | Full-context | RULER | HELMET | | Ours |
|---|---|---|---|---|---|---|
| | | | | Summ | LongQA | |
| | Qwen-2.5 0.5B | 16.7 | - | - | - | **20.6** |
| | Qwen-2.5 1.5B | 26.3 | - | 26.3 | **28.7** | 27.4 |
| | Qwen-2.5 3B | 29.5 | - | 29.5 | 29.5 | **30.0** |
| | Qwen-2.5 7B | 34.1 | 36.4 | 34.5 | **37.6** | 37.2 |
| | Qwen-2.5 14B | 35.7 | 35.6 | - | - | **37.4** |
| | Qwen-2.5 32B | 33.9 | 35.1 | - | - | **36.6** |
| GTE 1.5B | Qwen-2.5 72B | 32.5 | 32.5 | 35.0 | 35.0 | **36.3** |
| | Llama-3.2 1B | 17.7 | - | 24.6 | 24.6 | **25.8** |
| | Llama-3.2 3B | 28.7 | - | 28.7 | **31.1** | 30.3 |
| | Llama-3.1 8B | 33.3 | **34.9** | 34.9 | 34.0 | 34.5 |
| | Llama-3.3 70B | 31.9 | 33.2 | 35.8 | 33.2 | **35.9** |
| | ProLong 64K | 24.9 | - | - | - | **32.2** |
| | Qwen-2.5 0.5B | 17.3 | - | - | - | **21.3** |
| | Qwen-2.5 1.5B | 26.8 | - | 26.8 | 27.7 | **28.2** |
| | Qwen-2.5 3B | 30.2 | - | 30.2 | 30.2 | **32.7** |
| | Qwen-2.5 7B | 34.1 | 36.8 | 34.9 | **36.9** | 36.9 |
| | Qwen-2.5 14B | 35.7 | 35.4 | - | - | **36.2** |
| | Qwen-2.5 32B | 33.9 | 34.6 | - | - | **37.2** |
| GTE 7B | Qwen-2.5 72B | 32.5 | 32.5 | **35.9** | 35.9 | 35.3 |
| | Llama-3.2 1B | 17.7 | - | 24.9 | 24.9 | **25.4** |
| | Llama-3.2 3B | 28.7 | - | 28.7 | 29.7 | **31.4** |
| | Llama-3.1 8B | 33.3 | **35.1** | 35.1 | 33.7 | 33.7 |
| | Llama-3.3 70B | 31.9 | 34.4 | **35.8** | 34.4 | 33.3 |
| | ProLong 64K | 25.9 | - | - | - | **32.3** |

Table 1: Comparison of our method against the baselines on the SummHay dataset. We report average A3CU F1 scores across three sampled summaries. For the baselines, we only report scores for models with context length estimates previously reported in prior work. See Table 8 in Appendix for context window estimate used in each experiment.

| Retriever | Summarizer | Full-context | RULER | HELMET | | Ours |
|---|---|---|---|---|---|---|
| | | | | Summ | LongQA | |
| | Qwen-2.5-1M 7B | 32.1 | 33.3 | 32.1 | 32.1 | **33.6** |
| GTE 1.5B | Qwen-2.5-1M 14B | 35.6 | 35.6 | 35.6 | 35.6 | **37.4** |
| | ProLong 512K | 31.0 | - | 31.0 | 31.0 | **32.3** |
| | Qwen-2.5-1M 7B | 32.1 | **32.9** | 32.1 | 32.1 | **32.9** |
| GTE 7B | Qwen-2.5-1M 14B | 35.6 | 35.6 | 35.6 | 35.6 | **36.6** |
| | ProLong 512K | 31.0 | - | 31.0 | 31.0 | **32.5** |

Table 2: A comparison of our method against the baselines for very long-context LMs. Except for RULER on Qwen-2.5-1M 7B, all baselines estimate a full 128K context length.

## 5.2 Generalization to new models

In our LM panel, we included a mixture of Qwen, Llama, and Jamba models (§2.1). To test the generalization of our method to a new model class, we report the performance for the Phi-3 series (Abdin et al., 2024). In Table 3, we compare our proposed method with the baseline using GTE 1.5B and 7B retrievers. We find that RULER estimates perform the best and our method is a close second. In contrast to Table 1, the HELMET summarization estimate is better than its LongQA-based estimate, but both underperform our method.

| Retriever | Summarizer | Full-context | RULER | HELMET | | Ours |
| | | | | Summ | LongQA | |
| --- | --- | --- | --- | --- | --- | --- |
| | Phi-3 Mini | 11.0 | **30.6** | 30.4 | 30.4 | **30.6** |
| GTE 1.5B | Phi-3 Small | 27.8 | - | 31.1 | 30.3 | **31.9** |
| | Phi-3 Medium | 29.4 | **30.7** | 29.9 | 29.4 | **30.7** |
| | Phi-3 Mini | 11.0 | **29.9** | 28.3 | 28.3 | **29.9** |
| GTE 7B | Phi-3 Small | 27.8 | - | **32.4** | 30.6 | 31.5 |
| | Phi-3 Medium | 29.4 | **30.7** | 30.5 | 29.4 | 30.3 |

Table 3: A comparison of our method against the baselines for the Phi-3 series.

## 5.3 Effectiveness of system pooling

To test the effectiveness of pooling systems using MBR decoding (§2.2), we compared the pooled estimate of the system against two variants based on silver references from a single LM. We experiment with Qwen-2.5 72B and Llama-3.3 70B. In Table 4, we compare the effectiveness of silver summaries. Notably, we find that the Qwen 72B-based estimate fares better than both the Llama 70B-based and pooled estimates. This could be because Qwen-2.5 provides slightly full-context performance compared to Llama-3.3 70B (see Table 1).

| Summarizer | Silver Reference LM(s) | | |
| | System Pooling | Qwen 72B | Llama 70B |
| --- | --- | --- | --- |
| Qwen-2.5 0.5B | **21.3** | **21.3** | **21.3** |
| Qwen-2.5 1.5B | **28.2** | 27.7 | **28.2** |
| Qwen-2.5 3B | **32.7** | **32.7** | **32.7** |
| Qwen-2.5 7B | **36.9** | **36.9** | **36.9** |
| Qwen-2.5-1M 7B | 32.9 | **34.8** | 32.9 |
| Qwen-2.5 14B | 36.2 | 35.4 | **36.6** |
| Qwen-2.5-1M 14B | **36.6** | **36.6** | **36.6** |
| Qwen-2.5 32B | 37.2 | **37.8** | 37.2 |
| Qwen-2.5 72B | **35.3** | - | **35.3** |
| Llama-3.2 1B | 25.4 | **26.3** | 25.4 |
| Llama-3.2 3B | **31.8** | **31.8** | **31.8** |
| Llama-3.1 8B | 33.7 | 33.7 | **35.1** |
| Llama-3.3 70B | 33.3 | **34.7** | - |
| Phi-3 Mini | **29.9** | **29.9** | **29.9** |
| Phi-3 Small | 31.5 | **31.8** | 28.3 |
| Phi-3 Medium | 30.3 | **31.5** | 27.7 |
| ProLong 64K | 32.3 | **32.7** | 32.6 |
| ProLong 512K | **32.5** | 32.0 | **32.5** |

Table 4: A comparison of system pooling against Qwen and Llama-based silver references (GTE 7B retriever). We don't compute estimates for Qwen 2.5 72B based on Qwen 2.5 72B silver references (and similarly for Llama 3.3 70B).

Based on these results, we perform further analysis of our silver references in the pooling setup. In Table 5, we report the counts for how often each silver LM is chosen in the top-3 post-MBR decoding. The notable outliers here are Qwen-2.5 72B (picked least often) and Llama-3.3 70B (picked most often). This shows a potential limitation of our pooling-based estimate. Although MBR decoding allows us to make better use of the target summary space, it is possible that low-quality summaries in the pool could adversely impact the overall performance, albeit only by a small margin. An interesting future work direction would be to explore Best-of-N sampling as an alternative to MBR decoding.

| Silver Reference LM (full-context) | Count |
|---|---|
| Qwen-2.5 72B | 33 |
| Llama-3.3 70B | 79 |
| Jamba-1.5 Mini | 54 |
| Qwen-2.5-1M 14B | 51 |
| ProLong 512K | 59 |
| Total | 276 |

Table 5: Counts of silver summaries from individual LMs post-MBR decoding. We pick top-3 summaries per input, so a total of 276 summaries.

## 5.4 Effect of sample size

As we describe in §2.2, we sample a subset of the dataset before generating silver references using our LM panel. To understand the effect of this sample size, we compare various sampling ratios in Table 6. Our results show that even a very small sample (10% ≈ 9 examples) is sufficient for our estimation and shows superior performance to baselines.

| Model | 10% | 25% | 50% | 75% | 100% |
|---|---|---|---|---|---|
| Qwen-2.5 0.5B | 21.3 | 21.3 | 21.3 | 21.3 | 21.3 |
| Qwen-2.5 1.5B | 27.7 | 28.2 | 27.7 | 27.7 | 27.7 |
| Qwen-2.5 3B | 32.7 | 32.7 | 32.7 | 32.7 | 32.7 |
| Qwen-2.5 7B | 36.9 | 36.9 | 36.9 | 36.9 | 36.8 |
| Llama-3.2 1B | 26.3 | 25.4 | 25.8 | 25.8 | 25.8 |
| Llama-3.2 3B | 31.8 | 31.4 | 31.4 | 31.4 | 31.4 |
| Llama-3.1 8B | 34.6 | 33.7 | 34.6 | 35.1 | 35.1 |

Table 6: A comparison of our method at various dataset sampling ratios (GTE 7B retriever).

## 5.5 Performance & Efficiency

Small LMs tend to have very limited effective context windows; therefore, optimal RAG is necessary for improved task performance. For large LMs with longer effective context windows, optimal RAG can provide efficiency gains while maintaining or improving task performance.

**Performance:** Our results from Table 1, Table 2, and Table 3 show the effectiveness of our method in models ranging from 0.5B to 72B parameters. For a given downstream task, the user can pick a model size that is most suited to their computing budget. For example, Qwen-2.5 ≤ 7B can run on a single 48GB GPU, while larger models would require up to 4×48GB GPUs.

**Efficiency:** Compared to the baselines, our method often provides a significantly shorter context length estimate (see Table 8 in the Appendix §A.1). Therefore, the final summarization run on the full dataset is much more efficient with our method. However, we acknowledge that our method requires task-specific additional inference time compute to determine the optimal context length. Similar compute is also needed for benchmarks such as RULER and HELMET that compute task averages. In Table 6, we showed that our estimation requires a very small sample of the dataset, so the marginal cost of our method would be lower as the size of the dataset increases.

## 6 Conclusion & Future Work

In this work, we presented a methodology for estimating optimal context length for RAG-based summarization systems. Unlike traditional long-context benchmarks, our method

is geared to a specific downstream dataset and models the estimate as a function of the entire experimental configuration. We show the superior performance of our method across model classes and sizes. We show a generalization of our method to new model classes, as well as its effectiveness on models with very long context windows (>500K). In future work, we plan to apply our method to other tasks, such as open-domain multi-document QA and long-document summarization (Zhou et al., 2023). Previous work has also shown that the relative performance of long context and retrieval varies between examples (Karpinska et al., 2024; Pratapa & Mitamura, 2025), so another future direction is to identify the optimal retrieval context length for each example. Using open-weight models allowed us to analyze our method across various model sizes within a reasonable compute budget. We expect future work to expand our LM panel to include larger API-based models such as Gemini or GPT.

Another line of work studies input compression methods (Jiang et al., 2024; Xu et al., 2024) that fit long inputs to a fixed context length. Although these are a promising alternative to full-context setup, they may suffer irreversible information loss (Pratapa & Mitamura, 2025). In this paper, we focus on the strengths of RAG while taking advantage of the long-context reasoning capabilities of recent LMs. We leave the exploration of input compression with long-context methods to future work.

## Limitations

We discuss the limitations of our estimation method and the potential ways for future work to improve them. We use a silver panel consisting of medium-sized open-weight models (§2.1). This silver panel might not work as effectively to estimate the optimal context length for a much larger model such as Gemini-2.5 Pro. In such situations, we believe the silver panel should consist of models with similar capacity. Additionally, we use a full context setup to get silver summaries (§2.2) and this might not work as well if the inputs are much longer than the context windows supported by the LMs in our silver panel. An option is to perform RAG by retrieving tokens up to the LM's supported context window (similar to our approach, smaller Qwen models in §3.3). Our analysis in §5.3 also highlighted the limitations of system pooling, and future work could explore the use of Best-of-N sampling to improve the pooling mechanism. Finally, our method relies on the availability of at least a few examples from the downstream dataset and might not work well if this sample is not representative of the downstream task.

## Acknowledgments

We thank Amanda Bertsch and Kimihiro Hasegawa for helpful discussions and feedback on this work. We also thank anonymous reviewers for their constructive feedback during the review process. Adithya Pratapa was supported by an LTI Ph.D. fellowship.

## Ethics Statement

In this work, we limit our focus to the content selection evaluation of our summarization systems. However, we acknowledge that the factual accuracy of summaries is of great practical importance and point the reader to related work on hallucination in text summarization. We believe that our RAG-based estimation procedure does not increase the chances of possible hallucination in text summarization systems.

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

# A  Appendix

## A.1  Context length estimates

In Table 7, we list our models and the context length estimates from our baselines. In Table 8, we present our full set of results, including the standard deviation across the summaries of the three sampled systems summaries and the context length for each setting.

## A.2  Experiment details

### A.2.1  Dataset

We truncate the input documents to 128K tokens. We start by truncating the longest documents first. Due to slight differences in the tokenization methods between model classes, we calibrate the maximum number of summary tokens across models. We first get the 80th percentile of summary length (in NLTK tokens) and use the model-specific word-to-token ratio to set the max summary tokens.

We use the following prompt for the summarization task,

```
{document}

Question: {question}

Answer the question based on the provided document.
Be concise and directly address only the specific question asked.
Limit your response to a maximum of {num_words} words.
```

| Summarizer | Size | Supported | Estimated | | |
| --- | --- | --- | --- | --- | --- |
| | | | RULER | HELMET Summ | LongQA |
| Qwen-2.5 0.5B | 0.5B | 32,768 | - | - | - |
| Qwen-2.5 1.5B | 1.5B | 32,768 | - | 32,768 | 16,384 |
| Qwen-2.5 3B | 3B | 32,768 | - | 32,768 | 32,768 |
| Qwen-2.5 7B | 7B | 131,072 | 32,768 | 65,536 | 16,384 |
| Qwen-2.5-1M 7B | 7B | 1,010,000 | 65,536 | 131,072 | 131,072 |
| Qwen-2.5 14B | 14B | 131,072 | 65,536 | - | - |
| Qwen-2.5-1M 14B | 14B | 1,010,000 | 131,072 | 131,072 | 131,072 |
| Qwen-2.5 32B | 32B | 131,072 | 65,536 | - | - |
| Qwen-2.5 72B | 72B | 131,072 | 131,072 | 32,768 | 32,768 |
| Llama-3.2 1B | 1B | 131,072 | - | 32,768 | 32,768 |
| Llama-3.2 3B | 3B | 131,072 | - | 131,072 | 65,536 |
| Llama-3.1 8B | 8B | 131,072 | 32,768 | 32,768 | 65,536 |
| Llama-3.3 70B | 70B | 131,072 | 65,536 | 32,768 | 65,536 |
| ProLong 64K | 8B | 65,536 | - | - | - |
| ProLong 512K | 8B | 524,288 | - | 131,072 | 131,072 |
| Phi-3 Mini | 3B | 131,072 | 32,768 | 65,536 | 65,536 |
| Phi-3 Small | 7B | 131,072 | - | 32,768 | 65,536 |
| Phi-3 Medium | 14B | 131,072 | 32,768 | 65,536 | 131,072 |
| Jamba-1.5 Mini | 52B-A13B | 262,144 | - | 131,072 | 131,072 |

Table 7: A summary of LMs used in our work. We report the model size and context lengths (supported and estimated). For RULER and HELMET, we use the results reported in prior works to identify the context length estimates. Since our proposed context length estimate is also dependent on the retriever and dataset, we do not include those numbers here (see Table 8).

### A.2.2 Generation

For summary generation, we used temperature sampling (0.5) and generated three summaries for each input. All the results we report are the average scores across three runs. For the retrieval task, we limit the length of each document to 1024 tokens.

### A.2.3 Compute

We use a single L40S GPU for all our retrieval runs. For our summarization task, we use up to four L40S GPUs.

| Summarizer | Full-context | RULER | HELMET Summ | HELMET LongQA | Ours |
|---|---|---|---|---|---|
| **Retriever: GTE 1.5B** | | | | | |
| Qwen-2.5 0.5B | $16.7_{\pm1.5}$ (32K) | - | - | - | $\mathbf{20.6}_{\pm0.9}$ (8K) |
| Qwen-2.5 1.5B | $26.3_{\pm0.8}$ (32K) | - | $26.3_{\pm0.8}$ (32K) | $\mathbf{28.7}_{\pm1.2}$ (16K) | $27.4_{\pm1.1}$ (8K) |
| Qwen-2.5 3B | $29.5_{\pm0.2}$ (32K) | - | $29.5_{\pm0.2}$ (32K) | $29.5_{\pm0.2}$ (32K) | $\mathbf{30}_{\pm0.6}$ (8K) |
| Qwen-2.5 7B | $34.1_{\pm1.1}$ (128K) | $36.4_{\pm1.0}$ (32K) | $34.5_{\pm0.9}$ (64K) | $\mathbf{37.6}_{\pm0.3}$ (16K) | $37.2_{\pm0.9}$ (24K) |
| Qwen-2.5-1M 7B | $32.1_{\pm0.3}$ (128K) | $33.3_{\pm0.6}$ (64K) | $32.1_{\pm0.3}$ (128K) | $32.1_{\pm0.3}$ (128K) | $\mathbf{33.6}_{\pm0.4}$ (56K) |
| Qwen-2.5 14B | $35.7_{\pm0.6}$ (128K) | $35.6_{\pm0.7}$ (64K) | - | - | $\mathbf{37.4}_{\pm0.3}$ (24K) |
| Qwen-2.5-1M 14B | $35.6_{\pm1.2}$ (128K) | $35.6_{\pm1.2}$ (128K) | $35.6_{\pm1.2}$ (128K) | $35.6_{\pm1.2}$ (128K) | $\mathbf{37.4}_{\pm0.7}$ (24K) |
| Qwen-2.5 32B | $33.9_{\pm0.7}$ (128K) | $35.1_{\pm0.7}$ (64K) | - | - | $\mathbf{36.6}_{\pm0.6}$ (16K) |
| Qwen-2.5 72B | $32.5_{\pm0.5}$ (128K) | $32.5_{\pm0.5}$ (128K) | $35_{\pm0.8}$ (32K) | $35_{\pm0.8}$ (32K) | $\mathbf{36.3}_{\pm0.3}$ (24K) |
| Llama-3.2 1B | $17.7_{\pm0.2}$ (128K) | - | $24.6_{\pm0.6}$ (32K) | $24.6_{\pm0.6}$ (32K) | $\mathbf{25.8}_{\pm1.9}$ (8K) |
| Llama-3.2 3B | $28.7_{\pm1.4}$ (128K) | - | $28.7_{\pm1.4}$ (128K) | $\mathbf{31.1}_{\pm0.5}$ (64K) | $30.3_{\pm0.7}$ (56K) |
| Llama-3.1 8B | $33.3_{\pm0.9}$ (128K) | $34.9_{\pm0.8}$ (32K) | $\mathbf{34.9}_{\pm0.8}$ (32K) | $34_{\pm0.5}$ (64K) | $34.5_{\pm0.5}$ (40K) |
| Llama-3.3 70B | $31.9_{\pm0.8}$ (128K) | $33.2_{\pm0.4}$ (64K) | $35.8_{\pm0.2}$ (32K) | $33.2_{\pm0.4}$ (64K) | $\mathbf{35.9}_{\pm0.2}$ (40K) |
| ProLong 64K | $24.9_{\pm0.6}$ (64K) | - | - | - | $\mathbf{32.2}_{\pm0.4}$ (16K) |
| ProLong 512K | $31_{\pm0.8}$ (128K) | - | $31_{\pm0.8}$ (128K) | $31_{\pm0.8}$ (128K) | $\mathbf{32.3}_{\pm0.3}$ (48K) |
| Phi-3 Mini | $11_{\pm0.3}$ (128K) | $\mathbf{30.6}_{\pm0.5}$ (32K) | $30.4_{\pm0.1}$ (64K) | $30.4_{\pm0.1}$ (64K) | $\mathbf{30.6}_{\pm0.4}$ (16K) |
| Phi-3 Small | $27.8_{\pm1.3}$ (128K) | - | $31.1_{\pm0.1}$ (32K) | $30.3_{\pm0.9}$ (64K) | $\mathbf{31.9}_{\pm0.2}$ (48K) |
| Phi-3 Medium | $29.4_{\pm1.3}$ (128K) | $\mathbf{30.7}_{\pm1.2}$ (32K) | $29.9_{\pm0.1}$ (64K) | $29.4_{\pm1.3}$ (128K) | $\mathbf{30.7}_{\pm1.2}$ (32K) |
| **Retriever: GTE 7B** | | | | | |
| Qwen-2.5 0.5B | $17.3_{\pm0.4}$ (32K) | - | - | - | $\mathbf{21.3}_{\pm0.4}$ (8K) |
| Qwen-2.5 1.5B | $26.8_{\pm0.3}$ (32K) | - | $26.8_{\pm0.3}$ (32K) | $27.7_{\pm0.6}$ (16K) | $\mathbf{28.2}_{\pm0.6}$ (24K) |
| Qwen-2.5 3B | $30.2_{\pm0.2}$ (32K) | - | $30.2_{\pm0.2}$ (32K) | $30.2_{\pm0.2}$ (32K) | $\mathbf{32.7}_{\pm1.1}$ (16K) |
| Qwen-2.5 7B | $34.1_{\pm1.1}$ (128K) | $36.8_{\pm0.5}$ (32K) | $34.9_{\pm0.4}$ (64K) | $\mathbf{36.9}_{\pm1.2}$ (16K) | $\mathbf{36.9}_{\pm1.2}$ (16K) |
| Qwen-2.5-1M 7B | $32.1_{\pm0.3}$ (128K) | $\mathbf{32.9}_{\pm0.2}$ (64K) | $32.1_{\pm0.3}$ (128K) | $32.1_{\pm0.3}$ (128K) | $\mathbf{32.9}_{\pm0.2}$ (64K) |
| Qwen-2.5 14B | $35.7_{\pm0.6}$ (128K) | $35.4_{\pm0.9}$ (64K) | - | - | $\mathbf{36.2}_{\pm0.4}$ (16K) |
| Qwen-2.5-1M 14B | $35.6_{\pm1.2}$ (128K) | $35.6_{\pm1.2}$ (128K) | $35.6_{\pm1.2}$ (128K) | $35.6_{\pm1.2}$ (128K) | $\mathbf{36.6}_{\pm0.1}$ (48K) |
| Qwen-2.5 32B | $33.9_{\pm0.7}$ (128K) | $34.6_{\pm0.2}$ (64K) | - | - | $\mathbf{37.2}_{\pm0.7}$ (32K) |
| Qwen-2.5 72B | $32.5_{\pm0.5}$ (128K) | $32.5_{\pm0.5}$ (128K) | $35.9_{\pm0.4}$ (32K) | $35.9_{\pm0.4}$ (32K) | $35.3_{\pm0.1}$ (24K) |
| Llama-3.2 1B | $17.7_{\pm0.2}$ (128K) | - | $24.9_{\pm0.3}$ (32K) | $24.9_{\pm0.3}$ (32K) | $\mathbf{25.4}_{\pm0.7}$ (16K) |
| Llama-3.2 3B | $28.7_{\pm1.4}$ (128K) | - | $28.7_{\pm1.4}$ (128K) | $29.7_{\pm0.2}$ (64K) | $\mathbf{31.4}_{\pm0.5}$ (32K) |
| Llama-3.1 8B | $33.3_{\pm0.9}$ (128K) | $35.1_{\pm0.2}$ (32K) | $\mathbf{35.1}_{\pm0.2}$ (32K) | $33.7_{\pm0.4}$ (64K) | $33.7_{\pm0.4}$ (56K) |
| Llama-3.3 70B | $31.9_{\pm0.8}$ (128K) | $34.4_{\pm0.5}$ (64K) | $35.8_{\pm0.8}$ (32K) | $34.4_{\pm0.5}$ (64K) | $33.3_{\pm0.6}$ (80K) |
| ProLong 64K | $25.9_{\pm0.6}$ (64K) | - | - | - | $\mathbf{32.3}_{\pm0.7}$ (32K) |
| ProLong 512K | $31_{\pm0.8}$ (128K) | - | $31_{\pm0.8}$ (128K) | $31_{\pm0.8}$ (128K) | $\mathbf{32.5}_{\pm0.6}$ (32K) |
| Phi-3 Mini | $11_{\pm0.3}$ (128K) | $\mathbf{29.9}_{\pm0.4}$ (32K) | $28.3_{\pm1.4}$ (64K) | $28.3_{\pm1.4}$ (64K) | $\mathbf{29.9}_{\pm0.4}$ (32K) |
| Phi-3 Small | $27.8_{\pm1.3}$ (128K) | - | $\mathbf{32.4}_{\pm0.7}$ (32K) | $30.6_{\pm1.3}$ (64K) | $31.5_{\pm0.6}$ (24K) |
| Phi-3 Medium | $29.4_{\pm1.3}$ (128K) | $\mathbf{30.7}_{\pm0.9}$ (32K) | $30.5_{\pm0.5}$ (64K) | $29.4_{\pm1.3}$ (128K) | $30.3_{\pm1.4}$ (80K) |

Table 8: Full set of results on the SummHay dataset. For each system, we report the average score and standard deviation across three runs. We also provide the (optimal) context length estimate used for each experiment configuration in parantheses.

