# OpenReview forum: "Estimating Optimal Context Length for Hybrid Retrieval-augmented Multi-document Summarization"
_colmweb.org/COLM/2025/Conference — COLM 2025_

### Official Review · Reviewer_3Jr7 · 2025-05-05

**Rating:** 7
**Confidence:** 3
**Ethics Flag:** 1

**Summary:**

The paper addresses the task of Long Context Multi-Document summarization within a retrieval-augmented setting, where the retriever first selects a subset of the source documents which the LLM then uses to generate a query-focused summary. One of the main challenges in this setting is determining how much source content to retrieve to balance relevant content vs potential noise, given the available context window length of the summarizer LLM, and the paper introduces a novel approach for estimating an optimal retrieval size (in tokens), as a function of the retriever, dataset and summarizer. The idea is to use full-context LLMs to generate a pool of silver summaries for a sample of the dataset, and then to evaluate different RAG setups (retriever & summarizer) to determine the optimal context size. The experimental evaluation on the SummHay dataset covers an impressive range of Large and Long-Context LMs in various model sizes (0.5B - 72B), and the proposed approach shows strong performance across model classes, sizes, and retrievers.

**Quality/Significance:** This paper has a very high technical quality, with sound experimental setup and reporting (multiple sample generations, averaged results and variance, detailed analyses of the method in various settings). It presents a well-executed approach to a timely, relevant problem in the area of long context generation.

**Clarity:** The paper  well-structured and clearly written. The appendix provides the prompts used and more detailed experimental results (standard deviation, estimated context size)

**Originality:** I found the idea interesting and novel, but I might have missed relevant literature.

**Reasons To Accept:**

- Addresses an interesting and relevant problem in long context generative tasks
- Well-executed and clearly documented approach
- Impressive array of experiments / models analyzed, making for a convincing evaluation

**Reasons To Reject:**

- The experimental evaluation uses only a single dataset (I assume because no others are available), and focuses on the single task of summarization, which might limit the generalizability of the approach.

---

> ### Author Response · Authors · 2025-06-03
>
> We thank the reviewer for their encouraging comments. We are glad that they highlighted the technical quality, experiment setup and novelty of our method.
>
> > Single dataset
>
> We acknowledge the reviewer’s concern, but unfortunately, we couldn’t find another query-focused summarization dataset suitable for our setting. We are open to suggestions from the reviewer. We will be releasing our code under an open license, allowing us and others to extend our approach to new datasets.

---

> > ### Comment · Reviewer_3Jr7 · 2025-06-06
> > **Thank you**
> >
> > Thank you for the clarification, I'll keep my score as it is.

---

### Official Review · Reviewer_h4ue · 2025-05-10

**Rating:** 7
**Confidence:** 2
**Ethics Flag:** 1

**Summary:**

The paper proposes and validates a method to identify the context length as a function of a dataset, retriever, and summarizer combination in retrieval augmented generation. First a subset of pseudo examples is created from each dataset, then candidate silver summaries generated, then a single candidate summary selected, and finally a RAQ length chosen empirically by attempting to generate those silver summaries with different allowed context lengths.

**Questions To Authors:**

Line 53: I suggest the "representative examples" be clarified - examples of what? This underspecification is rather confusing.

Table 1: it is not immediately obvious on first read that the correct comparison is actually the last column against the first column, and the others are ablations.

Suggested related work:
- "Open Domain Multi-document Summarization: A Comprehensive Study of Model Brittleness under Retrieval
" https://arxiv.org/abs/2212.10526

Consider this dataset:
ODSum: New Benchmarks for Open Domain Multi-Document Summarization https://arxiv.org/abs/2309.08960

**Reasons To Accept:**

(A1) The paper provides a novel,  empirical, reference-free method for choosing context/budget size in RAG systems.

(A2) The method appears to perform well above a full-context baseline, and seems to generally perform similarly or better vs. ablations and other methods (Tables 1, 2), and adequate generalization performance to believe the result may be durable (Table 3)

**Reasons To Reject:**

(R1) The fundamental premise is not theoretically sound -- both the question and amount of literature within the dataset addressing the question are ignored. For some topics where a thorough review is required, the methodology suggested here will continue to work, but for others (e.g. in medical literature reviewing), the recall burden is substantially higher.

(R2) The actual description of the method is not particularly clear and should be revisited to be unambiguous: exactly what is sampled from the dataset (an "example" could be a set of documents with arbitrary relation, it could be a Q/A pair, or something else), how is it sampled ("random" is not a very clear description),

---

> ### Author Response · Authors · 2025-06-03
>
> We thank the reviewer for their thoughtful comments. We are glad they highlighted the novelty of our method and the strengths of our experimental setup.
>
> > The fundamental premise is not theoretically sound -- both the question and amount of literature within the dataset addressing the question are ignored.
>
> This is an interesting point. To some extent, we agree with the reviewer’s comment about the impact of “amount of literature within the dataset addressing the question”. This is a critical factor in determining how much to retrieve. However, we want to note that unlike the baselines that rely solely on the LLM, our method does consider the impact of the dataset when determining the optimal retrieval length. So, for datasets that require more detailed review (i.e., information is more spread), our method will find a longer retrieval length.
>
> Additionally, iterative RAG methods could be useful tools in these settings. As we mention in lines 167-176, they have previously been explored for QA tasks. Our methodology could be integrated into iterative RAG methods by facilitating the selection of optimal retrieval length at each iteration.
>
> > Description of our method and clarification about representative examples
>
> Thanks for the suggestions, we will make changes to our paper to clarify these. We use an example to indicate an input-output pair, where input consists of a set of documents (+ a query) and output is the target summary. We sample a fraction of these examples to generate silver target summaries. We fix the random seed to allow for reproducibility.
>
> > Table 1 comparisons
>
> The RULER and HELMET columns constitute our baselines. As we describe in section 2.1, we use these existing long-context benchmarks to get optimal context window estimates for our LLMs.
>
> > Related work: Open Domain Multi-document Summarization: A Comprehensive Study of Model Brittleness under Retrieval
>
> Thanks for the pointer! We will include this in our related work.
>
> > Dataset suggestion: ODSum: New Benchmarks for Open Domain Multi-Document Summarization
>
> Thanks for the suggestion, this dataset looks interesting. This dataset differs from SummHay (used in our paper) in terms of the task setting. ODSum uses an open-domain MDS setting where the system is required to perform retrieval over a large document index for each query. While this is an interesting setting, it’s not clear if a full-context baseline can be used for open-domain MDS. In our approach, we use a full-context setup both as a baseline and to generate silver summaries using larger LMs.
>
> One potential way to use ODSum is to approximate retrieval@128k as the full-context baseline. We will try this option in the next draft.

---

> > ### Comment · Reviewer_h4ue · 2025-06-05
> >
> > Can you add more of a description here of the actual method so I can better understand it? The method is the _key contribution_ of the paper, and is it is unclear to this reviewer. It's hard to assess a method/method's description without reading it.

---

> > > ### Author Response · Authors · 2025-06-06
> > >
> > > Definitely! Our goal is to decide how many relevant documents to retrieve before generating a summary. For a given dataset (D), retriever (R) and summarizer (S), our method has the following steps.
> > >
> > > 1. We sample a subset of examples from D. Each example in this subset constitutes a set of documents and a query.
> > > 2. We use a panel of LLMs to generate summaries for this subset. These summaries serve as our silver references.
> > > 3. We then use Minimum Bayes Risk decoding to identify the top silver references.
> > > 4. We conduct a search over retrieval lengths (8k to 80k) by comparing the system-generated summary (using R & S) against the silver references. This search gives us the optimal retrieval length estimate for our RAG setup.
> > > 5. Finally, on the full dataset, we retrieve the top-k documents that fit into this length estimate (using R) before generating a summary (using S).
> > >
> > > Our baselines use the optimal context lengths estimates for the summarizers (S) from two long-context benchmarks, RULER and HELMET.
> > >
> > > We hope this clarifies our method. We are happy to answer any follow up questions.

---

> > > > ### Comment · Reviewer_h4ue · 2025-06-06
> > > >
> > > > I have updated my score on the promise that these details will appear, but:
> > > >
> > > > (1) the exact method of sampling should be unambiguously described (uniform? some other method?). "Random" is not a precise description of a sampling method.
> > > >
> > > > (2) the paper should be unambiguous about what exactly is sampled; what comprises can instance. The reply clarifies this greatly, and this should be a minor change to the actual prose.
> > > >
> > > > (3) this reply misunderstands my concern (R1): multiple documents may appear to answer a question, they may contradict each other, this contradiction might be correct/true/due to randomness in the world, thus answering this class of question requires both a high recall and high quality analysis. This method seems more targeted to questions where an answer is present in one document or by hops through several documents. I understand that these datasets are relatively uncommon, so while this is a weakness of the method, the community seems less interested in this type of synthesis problem.

---

### Official Review · Reviewer_iKj6 · 2025-05-10

**Rating:** 6
**Confidence:** 4
**Ethics Flag:** 1

**Summary:**

The paper proposes an approach to estimate optimal input context lengths to be retrieved by a RAG for multi-document summarisation. First, a subset of the training dataset is sampled. A panel of LLMs (Llama, Qwen and Jamba) is used to generate a pool of silver-references for the sampled dataset. Then, the system is tested over the sample data (RAG+summarizer), with a varying window of context lengths from 8 to 80K in 8K intervals, generating three predictions per input, and evaluated against the silver reference pool. The smallest context length that falls within a standard deviation of the maximum score is selected as the optimal length. Finally, once the optimal lengths is selected for a particular dataset, RAG model and summarizer, the authors test the system on the test dataset. Overall, their test  results generally outperform the comparing baselines with full-context models and context-lengths optimised on the RULER and HELMET benchmarks, with generally shorter estimated context lengths, which make their approach more efficient at inference time. Additionally, the paper presents a sensitivity analysis on various parameters in their approach such as the percentage of samples used to estimate the context lengths, the use of LLMs with large context windows, and the use of LLMs from a different family compared to the ones used to create the pool of silver references.

**Questions To Authors:**

- I suggest to include the estimated context lengths in table 1 for each baseline and the proposed approach. It is an important information to understand the contribution, it should be included in the main paper, not just the Appendix.

**Reasons To Accept:**

- The paper is well written and the main contribution clearly presented.
- Interesting experimental results, demonstrating competitive performance compared to other strong baselines (outperforming the baselines in most settings) with generally shorter context lengths, which make their approach more efficient at inference time.
- The approach seems to generalise well across various LLMs and RAG models.

**Reasons To Reject:**

- The experimental analysis has been carried out over a single summarisation dataset. The authors claim that the optimal context-length depends on the dataset, RAG and summarizer. However, they don't explore if their context-length estimation works in other datasets. It would be interesting to understand if the approach works well with different tasks too, not only summarisation (although authors have already mentioned that they are planning to explore this in future work).
- The RULER and HELMET baselines seem to be at a disadvantage, because their optimal  context-length is obtained over different datasets, not the sample obtained from the SummHay dataset. It would be interesting to compare the authors' approach with a simple baseline that uses the same sample dataset (potentially even with the respective ground-truth references) and the coarser varying input lengths proposed in the baselines.

---

> ### Author Response · Authors · 2025-06-03
>
> We thank the reviewer for their encouraging comments. We are glad that the reviewer highlighted the strength of our experimental setup and the generalizability of our method. We respond to their concerns and questions below,
>
> > Single summarization dataset
>
> We acknowledge the reviewer’s concern, but unfortunately, we couldn’t find another query-focused summarization dataset suitable for our setting. We are open to suggestions from the reviewer. We will be releasing our code under an open license, allowing us and others to extend our approach to new datasets.
>
> > RULER and HELMET baselines seem to be at a disadvantage
>
> We understand the reviewer’s concern about the differences in datasets used in these benchmarks vs our setup. However, our goal in comparing RULER and HELMET is to check if an LLM-specific context window estimate is transferable to downstream tasks (custom retrievers and datasets). While these baselines provide a holistic view of the LLM performance (especially HELMET), our results show that they fall short on new datasets. We like the reviewer's suggestion of using coarser input lengths for a fair comparison. We will add these results to our paper. We didn’t include an experiment that uses ground-truth references from SummHay because this would mean we have to exhaustively search over all context windows using the test labels. This is often infeasible in real-world RAG use-cases where the ground-truth references are unavailable.
>
> > Estimated context lengths in Table 1
>
> Thanks for the suggestion! We will include them in Table 1.

---

### Official Review · Reviewer_jEKZ · 2025-05-12

**Rating:** 7
**Confidence:** 3
**Ethics Flag:** 1

**Summary:**

This paper describes a multi-document summarization proposal that searches for a good context length, uses it to decide how many relevant documents to retrieve, before generating a summary based on the retrieved docs.

The first step involves sampling a subset of the set of documents to be summarized. These are passed through a panel of different large-language models (LLMs) (e.g. Qwen, Llama, etc) to obtain a set of 'silver' reference summaries. Minimum Bayes Risk (MBR) decoding is then used to identify the top silver summaries. After this, a search is conducted over a fairly wide search space of varying context lengths by comparing outputs from a summarizer with the top set of silver summaries to identify what context length works best. With this estimated context length, top k documents which can fit into this length are retrieved, and used as input to the final summarizer step.

Evaluations are done over the SummHay dataset, against baselines including 1/ one using the full context window of a summarizer, as well as previously reported runs for 2/ RULER, and 3/ HELMET on the dataset. Results generally show better A3CU F1 scores when smaller LLM summarizers are used. The effect is stronger when paired with a relatively larger embedding model. Further experiments conducted over LLMs. with very long context windows suggests that this proposal works well too.

**Questions To Authors:**

Suggestion:

1. It takes a few reads to properly piece together what the authors are proposing. There are attempts to explain what the proposed system is comprised of at various parts of the doc. With each attempt, there is progressively more context and details that are given. However, this means that each time, the reader has to pause, do a mental diff of the relatively more vague explanation they read earlier, and reconcile it with the more detailed explanation they have in front of them. After a while, it gets tedious. It would help to focus the explanation in a methodology section and give the reader a thorough read without having to build up a series of diffs over the course of several pages.

**Reasons To Accept:**

1. Good set of experiments conducted, they build a good case for the efficacy of the approach - starting from baseline comparisons, to exploring different setups and baselines (e.g. against very long context LLMs, against new LLMs, etc)
2. For a given RAG-based multi-doc summarization setup, the proposal has the potential to produce cost savings through the use of more optimal choice of context lengths. This can be significant over time.

**Reasons To Reject:**

1. The paper lacks a clear motivation or value prop. My best guess right now is that the proposal could lead to cost savings through more optimal choices of context lengths. Yet the effect is more noticeable when a relatively larger embedding model is used for retrieval. It's not clear to me how the relative costs increase/decreases interact and whether this could be a nett gain.
2. Re the generation of silver summaries used to estimate the right context width, a natural question that brings about is how these might compare against the full RAG-based pipeline that are used in the experiment.
3. It took me several passes to get a good sense of what the proposal is (see suggestion #1). The paper could benefit from more editorial love.

---

> ### Author Response · Authors · 2025-06-03
>
> We thank the reviewer for their thoughtful comments, and we will take your suggestions to improve the presentation of our paper. We are glad the reviewer highlighted the strength of our experimental setup, and the potential of our method to reduce inference costs over time. We respond to their concerns and questions below,
>
> > Motivation and value proposition
>
> We want to highlight value propositions in terms of efficiency and performance. Some of these were already highlighted by the reviewer.
>
> Efficiency: There are two key comparisons here, 1. Full-context vs RAG, 2. Across RAG systems. RAG is often much more efficient than the full-context setup. We will update our paper to include average run times (incl. the cost of the retrieval step). Compared to baseline RAG setups, our method (often) picks the shortest retrieval length (Table 8). This leads to faster summary generation with the LLM. A large embedding model does increase the cost, but this cost is the same across RULER, HELMET and our estimation method.
>
> Performance: Prior work (Jin et al., 2025) and our work showed that the LLM’s performance at their max context window (here 128k) is much lower than RAG performance at a shorter context window. Additionally, our method provides a length estimate that improves upon the strong baselines (RULER, HELMET).
>
> > Silver summary generation and RAG-based pipeline
>
> For silver summary generation, we decided to use a full-context setup instead of a RAG setup for two reasons. First, as noted by the prior works (Jin et al., 2025) and our preliminary analysis, we find the RAG setup is highly sensitive to the choice of context length. We are focused on fixing this exact problem, therefore, we didn’t want to make a choice of context window during silver summary generation. Second, larger LMs tend to be better at longer contexts than smaller LMs, so we mostly limit our silver LLM panel to large models. Since we run the silver summary generation on a small subset, the cost of this (full-context) run is small.
>
> If we interpreted your comment incorrectly, please let us know!
>
> > Editing
>
> Thank you for putting in the effort to understand our work. We will improve the writing by describing our method in detail at the start of section 2 before diving into implementation details. We will also make more references to our Figure 1 to improve the readability.

---

> > ### Comment · Reviewer_jEKZ · 2025-06-09
> >
> > Thank you for your responses. I have gone through them and they helped me better understand the paper. It is a good paper, I appreciate your efforts! Thank you!

---

### Decision · Program_Chairs · 2025-07-08

**Decision:**

Accept

**Comment:**

This paper estimates the optimal input context length to be retrieved by a RAG system for multi-document summarisation. First, a subset of the training dataset is sampled. A panel of LLMs (Llama, Qwen and Jamba) is used to generate a pool of silver-standard references for the sampled dataset. Then, the system is tested over the sample data (RAG+summarizer), with a varying window of context lengths from 8 to 80K in 8K intervals, generating three predictions per input, and evaluated against the silver reference pool. The smallest context length that falls within a standard deviation of the maximum score is selected as the optimal length. Overall, their test results generally outperform comparison systems with full-context models and context-lengths optimised on the RULER and HELMET benchmarks, with generally shorter estimated context lengths, which make their approach more efficient at inference time. Additionally, the paper presents a sensitivity analysis on various parameters such as the percentage of samples used to estimate the context length, the use of LLMs with large context windows, and the use of LLMs from a different family compared to the ones used to create the pool of silver references.

Reviewers were broadly positive, and the authors clarified questions and misunderstandings in the rebuttal period. It would be nice to see whether using the gold summary instead of the silver-standard ones leads to different length estimates and different/better summarization results.